# Oral Bioavailability Evaluation of Celastrol-Encapsulated Silk Fibroin Nanoparticles Using an Optimized LC-MS/MS Method

**DOI:** 10.3390/molecules25153422

**Published:** 2020-07-28

**Authors:** Shuyu Zhan, Amy Paik, Felicia Onyeabor, Baoyue Ding, Sunil Prabhu, Jeffrey Wang

**Affiliations:** 1Department of Pharmacy, College of Medicine, Jiaxing University, Jiaxing 314001, China; lena_310@zjxu.edu.cn; 2Department of Pharmaceutical Sciences, College of Pharmacy, Western University of Health Sciences, Pomona, CA 91766, USA; amy.paik@westernu.edu (A.P.); felicia.onyeabor@westernu.edu (F.O.); sprabhu@westernu.edu (S.P.)

**Keywords:** LC-MS/MS, celastrol, silk fibroin nanoparticles, pharmacokinetics, bioavailability

## Abstract

Celastrol (CL), a compound isolated from *Tripterygium wilfordii*, possesses various bioactivities such as antitumor, anti-inflammatory and anti-obesity effects. In previous studies, we developed CL-encapsulated silk fibroin nanoparticles (CL-SFNP) with satisfactory formulation properties and in vitro cancer cytotoxicity effect. For further in vivo oral bioavailability evaluation, in this study, a simple and reliable LC-MS/MS method was optimized and validated to determine CL concentration in rat plasma. The separation of CL was performed on a C18 column (150 by 2 mm, 5 µm) following sample preparation using liquid–liquid extraction with the optimized extraction solvent of *tert*-butyl methylether. The assay exhibited a good linearity in the concentration range of 0.5–500 ng/mL with the lower limit of quantification (LLOQ) of 0.5 ng/mL. The method was validated to meet the requirements for bioassay with accuracy of 91.1–110.0%, precision (RSD%) less than 9.1%, extraction recovery of 63.5–74.7% and matrix effect of 87.3–101.2%. The developed method was successfully applied to the oral bioavailability evaluation of CL-SFNP. The pharmacokinetic results indicated the AUC_0-∞_ values of CL were both significantly (*p* < 0.05) higher than those for pure CL after intravenous (IV) or oral (PO) administration of equivalent CL in rats. The oral absolute bioavailability (*F*, %) of CL significantly (*p* < 0.05) increased from 3.14% for pure CL to 7.56% for CL-SFNP after dosage normalization. This study provides valuable information for future CL product development.

## 1. Introduction

Celastrol (CL, Figure 1A), a quinone methide triterpenoid isolated from *Tripterygium wilfordii*, possesses various bioactivities such as antitumor, anti-inflammatory and anti-obesity effects, and have been demonstrated to have treatment potential on multiple cancers [1], autoimmune diseases [2], chronic inflammation [3], cardiovascular diseases [4] and neurodegenerative diseases [5]. However, the poor aqueous solubility, low therapeutic index and systemic toxicity of CL limited its clinical applications [3,6]. To overcome CL’s defect as a prospective product in clinic, novel drug delivery systems of CL based on nanotechnology, including nanofibers [7], liposomes [8,9], nanomicelles [10,11] and various kinds of nanoparticles [12,13,14,15,16] using single or synergistic delivery strategy, were designed and evaluated. The clinical application of CL is limited mainly due to its unsatisfactory in vivo pharmacokinetic as well as pharmacodynamic profiles. Recently, we developed a celastrol-encapsulated silk fibroin nanoparticle (CL-SFNP) using a desolvation method [17,18]. In vitro drug release studies of CL-SFNP revealed a slow and sustained release of drug at physiological pH (pH 7.4) and rapid release at lysosomal pH (pH 4.5), and the in vitro cytotoxicity study against two human pancreatic cancer cell lines (PANC-1 and Mia PaCa-2) demonstrated increased growth inhibition with the nanoparticle formulation compared to pure CL. Therefore, it is necessary to further study this formulation’s in vivo properties for possible product development of CL.

Pre-clinical pharmacokinetic evaluation including oral bioavailability is a vital part of the drug development process, and provides valuable information for the drug candidate’s successful clinical translation. Nevertheless, the reliable determination methods are essential for the precise pharmacokinetics of the analyte and its novel formulations. Till now, some determination methods of CL in plasma based on HPLC technology and their application in pharmacokinetic studies have been reported [18,19,20,21]. However, the insufficient selectivity and sensitivity of HPLC methods apparently limit their wide application on in vivo pharmacokinetics such as oral bioavailability studies of CL loaded drug delivery systems. Zhang et al., reported an LC-MS/MS method for determination of CL in rat plasma, however, it needed complicated mobile phase, larger sample volume of 100 μL and time-consuming sample preparation [22]. Likewise, no study has been reported to investigate CL’s plasma sample preparation method concerning its high hydrophobic property leading to lower extraction recoveries of about 50% [18,22]. Therefore, it is necessary to develop a simple and reliable determination method of CL in plasma with optimized sample preparation. In this study, in order to evaluate CL-SFNP’s pre-clinical pharmacokinetics and bioavailability, we tried to develop a reliable LC-MS/MS method to determine CL in rat plasma coupled with simple and effective sample preparation.

## 2. Materials and Methods

### 2.1. Chemicals, Reagents and Materials

CL standard was purchased from Medkoo Biosciences Inc. (Morrisville, NC, USA). 18-α Glycyrrhetnic acid (Internal Standard, IS, Figure 1B) and HPLC-grade formic acid were bought from Sigma (St. Louis, MO, USA). Rat plasma was purchased from Innovative Research (Novi, MI, USA). HPLC-grade acetonitrile and methanol were purchased from VWR (Randor, PA, USA). Ultrapure water was produced by Millipore equipment (Millipore, Bedford, MA, USA). All other reagents were obtained from commercial sources and were of analytical grade.

### 2.2. Chromatographic and Mass Spectrometry Conditions

The LC-MS/MS system was composed of a SHIMADZU LC-20AD liquid chromatography system equipped with a SIL-20A auto sampler, and a triple-quadrupole tandem mass spectrometer (API 3200, Applied Biosystems, Foster City, CA, USA). The chromatographic separation was achieved on a Phenomenex Gemini C18 column (150 by 2 mm, 5 µm). The mobile phase consisted of 0.1% formic acid-2 mM ammonium acetate aqueous solution (A) and acetonitrile (B) using a gradient elution of 50% B at 0–2 min, 50–95% B at 2–3 min, 95% B at 3–6 min, 95–50% B at 6–7 min and the stop time was at 10 min. The flow rate was 0.35 mL/min. The sample injection volume was 10 µL. CL and IS were both determined using positive ionization. The analytes were quantified under the multiple-reaction monitoring (MRM) mode. Mass spectrometry was operated with an optimized ion spray voltage at 3000 V, turbo spray temperature at 500 °C, and collision gas at 5 psi. The curtain gas, nebulizer gas (Gas 1) and auxiliary gas (Gas 2) were at 25, 40 and 20 psi, respectively. The precursor-to-product ion pair of m/z 451.193→201.096 for CL and m/z 471.293→135.178 for IS were monitored with the optimized declustering potential (DP), entrance potential (EP), collision energy (CE) and collision cell exit potential (CXP) of 31, 7, 33 and 4 V and 81, 12, 33 and 4 V, respectively.

### 2.3. Preparation of Calibration Standards and Quality Control Samples

The stock solution of CL (1.0000 mg/mL) was prepared in methanol. A series of standard working solutions with the concentrations of 500, 200, 100, 50, 25, 5, 1 and 0.5 ng/mL were obtained by diluting the stock solution with methanol. The IS stock solution (0.5 mg/mL) was also prepared in the methanol. All solutions were stored at 4 °C. To prepare calibration curves, each standard working solution (50 μL) was evaporated to dryness in an Eppendorf Vacufuge Plus (Eppendorf, Germany). The residue was then spiked with 50 μL blank rat plasma and mixed to form calibration standards. To validate the method, high, middle, and low quality control (QC) samples containing CL of 200, 25 and 1 ng/mL, respectively, were prepared with the same treatment as the calibration standard samples. The calibration standard and QC samples were prepared for each analysis batch.

### 2.4. Sample Preparation

For CL analysis in rat plasma, the plasma samples were extracted using *tert*-butyl methylether based on liquid-liquid extraction technique. To each tube containing 50 μL plasma, 1.0 mL *tert*-butyl methylether containing 1.0 μg/mL IS was added. The mixture was vortex-mixed for 5 min at 3000 rpm using a VWR Pulsing Vortex Mixer (Randor, PA, USA) and centrifuged for 5 min at 10,000 rpm. Then, 900 μL of supernatant was removed and placed into a new tube and evaporated at 30 °C to dryness in the Eppendorf Vacufuge Plus (Eppendorf, Hamburg, Germany). The residue was reconstituted in 100 μL methanol and centrifuged for 5 min at 10,000 rpm, and a 10.0 μL aliquot was injected for the LC-MS/MS analysis within 18 h.

### 2.5. Method Validation

#### 2.5.1. Specificity

The specificity of LC-MS/MS method was evaluated by comparing the MRM chromatograms of blank rat plasma, blank rat plasma spiked with CL and IS, and rat plasmas collected at 0.5 h after drug administration. No observation of endogenous interference in the MRM chromatogram for CL and IS analysis was used to confirm the method’s specificity.

#### 2.5.2. Linearity and Low Limit of Quantification

The calibration curves were prepared by assaying standard plasma samples as described in preparation of calibration standards and quality control samples, and each level sample was prepared and assayed in duplicate in three separate days. The calibration curves were plotted by the peak area ratio (analyte/IS, y) versus the analyte concentration (x) using a 1/x^2^ weighted linear least-squares regression model. The lower limit of quantification (LLOQ) was defined as the lowest concentration on the calibration curve that produced a signal-to-noise (S/N) ratio of at least 10 with an acceptable accuracy (recovery within 80–120%) and precision (RSD, below 20%).

#### 2.5.3. Precision and Accuracy

The precision and accuracy were evaluated by analyzing six replicates of QC samples at low, medium and high concentration levels as described in preparation of calibration standards and quality control samples. The intra-day precision and accuracy of the assays were determined in the same day by analyzing six replicates at each concentration level. The inter-day precision and accuracy were evaluated on six consecutive days at each concentration level. The precision values expressed as RSD were required to be below 15%, and the accuracy values expressed by recovery (mean measured concentration/spiked concentration × 100%) were required to be within 85–115%.

#### 2.5.4. Extraction Recovery and Matrix Effect

The extraction recoveries were determined by comparing the peak area ratio (analyte/IS) from blank plasma samples spiked before extraction with those from blank plasma samples spiked after extraction at three levels of QC samples with IS (*n* = 6). The matrix effect at three levels of QC samples with IS (*n* = 6) was investigated by comparing the peak area ratios (analyte/IS) of CL and IS spiked with the blank plasma samples after extraction to those of pure standard solutions containing CL and IS at the same concentrations.

#### 2.5.5. Stability

The stability of CL in plasma was assessed by analyzing six replicates which were spiked with standards at low, medium and high QC samples under different conditions: In autosampler after preparation for 18 h, at −20 °C for 30 days and three freeze-thaw cycles. Newly prepared calibration curve was used for every stability test. The stability was evaluated by deviation, (spiked concentration—mean measured concentration)/(spiked concentration) × 100%, which was required to be within ±15%.

### 2.6. Pharmacokinetic Studies

The study protocol (R18IACUC001) was approved by Western University of Health Sciences Institutional Animal Care and Use Committee (IACUC). Male Sprague–Dawley rats with pre-cannulated jugular vein were separated into four groups (*n* = 3). Two groups were used for intravenous (IV) administration study and the other two were used for oral (PO) administration study. The drug solutions of administration were prepared as pure CL solution in PEG 300 and CL-SFNP suspension in phosphate buffer solution (PBS), pH 7.4 respectively. The rats were administered intravenously and intragastrically with the CL dose of 1 mg/kg and 3 mg/kg, respectively. Two hundred μL of blood was collected respectively into pre-heparinized tubes at 0.083, 0.5, 1, 2, 4, 6, 8, 12 and 24 h after IV administration, and at 0.25, 0.5, 0.667, 1, 2, 4, 6, 8, 12 and 24 h after PO administration. After 24 h the rats were euthanized with 30% isoflurane. Collected blood was then centrifuged for 15 min at 14,000 rpm. After centrifugation 110 μL plasma was collected and transferred into an Eppendorf tube, and then stored at −20 °C until analysis. The IV samples were initially analyzed using an HPLC-UV method [18], but the low sensitivity of the method prohibited the analysis of the oral samples.

### 2.7. Data Analysis

Data were expressed as the mean ± standard deviation (SD). The weighted linear least-squares regression model was performed using SPSS 16 software. The major pharmacokinetic parameters were estimated by non-compartmental analysis using WinNonlin 5.3 program package (Certara USA, Inc., Saint Louis, MO, USA). Differences between two groups were evaluated by an unpaired Student’s *t*-test using Excel Statistical function (Microsoft, Redmond, WA, USA), and three groups were evaluated by one-way ANOVA using SPSS 16 software (IBM, Armonk, NY, USA). Differences were considered statistically significant at *p* < 0.05.

## 3. Results and Discussion

### 3.1. LC-MS/MS Method

In the LC-MS/MS method development, chromatographic condition was optimized, and the mobile phase with the additive of 0.1% formic acid-2 mM ammonium acetate in aqueous solution (A) and a gradient elution was chosen, which resulted in good separation and peak shapes of CL and IS. Owing to CL’s high non-polarity, higher ratio of organic mobile phase (B) is usually needed to reduce elution time. However, it was found that high ratio (90%) of organic mobile phase with an isocratic elution, causing CL’s retention time at about 3 min, could cause not only poor separation of CL from IS but also very high matrix effect (>200%) from plasma especially for CL analysis at low concentrations. Therefore, to mediate the contradiction between elution time and separation as well as to reduce matrix effect, we applied a gradient elution of 50% B at 0–2 min, 50–95% B at 2–3 min, 95% B at 3–6 min, 95–50% B at 6–7 min with the total analysis time of 10 min, and injected the eluent into the MS system at 5.5 min using the valve as a diverter. As a result, the suitable chromatographic separation of CL and IS with the retention times of 7.31 min and 6.35 min respectively was obtained and the method exhibited no apparent matrix effect from plasma.

### 3.2. Optimization of Sample Preparation

Liquid–liquid extraction is the most popular plasma sample preparation technique because of its simple, convenient and reproducible procedure [23,24,25,26]. However, chaotic extraction recoveries of CL from plasma based on LLE have been reported, from about 60% to 90% using ethyl acetate as extract solvent with one-step extraction [27,28], from about 50% to 90% using acetonitrile as extract solvent with one-step extraction [18,29], and about 50% using trichloromethane as extract solvent with two-step extraction [22]. In this study, we tried to optimize extract solvent to improve extraction recovery by investigating a wide-range of organic solvents of acetonitrile, methanol, ethyl acetate, 2-butanone, hexane, ethyl butrate, iso-propyl ether, *tert*-butyl methylether, dichloromethane and trichloromethane. To 50 μL blank plasma sample spiked with 100 ng/mL CL was added 1 mL extract solvent, and then was processed according to the method of sample preparation. Each solvent was investigated with three replicates. The peak area of CL obtained from each extract solvent is illustrated in Figure 2. It showed that *tert*-butyl methylether extracted the highest quantity of CL from plasma among all investigated solvents, so it was chosen as the extract solvent in our method. Using the optimized extract solvent, we could obtain extraction recovery of about 70% which was higher than some of those reported [18,22]. To test if increase of time and number of extractions would improve extraction efficiency, one-step extraction with 5 min, one-step extraction with 10 min and two-step extraction with each 5 min were further investigated at three levels of QC samples with IS (*n* = 6). The peak area ratios (CL/IS) obtained from different extract conditions are listed in Table 1. It shows, for all three concentration levels of CL, no significant difference (*p* > 0.05) of peak area ratios existed among different extract conditions. Taken together, one-step extraction with 5 min was chosen as the simple and time-saving extraction method with recovery of about 70% in sample preparation. Incomplete extraction of CL from plasma might be due to its high hydrophobicity or partially irreversible plasma protein binding, which needs to be further investigated.

### 3.3. Method Validation

#### 3.3.1. Specificity

The representative chromatograms of a blank plasma sample, a blank plasma sample spiked with CL and IS, and two plasma samples obtained at 0.5 h after IV and PO administration of CL-SFNP suspension are shown in Figure 3. No significant interferences from endogenous substances in the blank plasma sample were observed at the retention times of CL and IS. Good specificity and selectivity were thus observed for the method.

#### 3.3.2. Linearity and LLOQ

The linearity was validated in CL concentration range of 0.5–500 ng/mL by determining eight different concentration calibration standards in three separate days. Good linearity was obtained with a typical linear regression equation of y = 0.004x (*r*^2^ ≥ 0.976) using 1/x^2^ weighting. LLOQ was 0.5 ng/mL with recovery of 91.1–112.5% and RSD of 11.2%. Although the sensitivity with LLOQ of 0.5 ng/mL was a little lower than that reported (0.1 ng/mL) [22], it is meaningful for our method because we used a smaller plasma sample volume of 50 μL, which would add benefit for multiple time points pharmacokinetic studies.

#### 3.3.3. Precision and Accuracy

The intra-day and inter-day precision and accuracy of the method were determined by replicating analytes of QC samples at 1, 25 and 200 ng/mL concentration levels. The results are listed in Table 2. The mean accuracy of the analyte was within the range of 91.1–110.0%. The intra- and inter-day precisions (RSD %) of these analytes were less than 9.1% and 11.7%, respectively. The results demonstrated the method developed in this study was precise and accurate.

#### 3.3.4. Extraction Recovery and Matrix Effect

The extraction recovery and matrix effect of CL in rat plasma are shown in Table 3. It shows relatively high extraction recovery at three concentration levels with the mean values of 67.0%, 63.5% and 74.7%. The matrix effects of CL at three concentration levels were found to be within the range of 87.3–101.2%, demonstrating no apparent influence of rat plasma matrix on CL determination.

#### 3.3.5. Stability

Results from stability studies of CL are shown in Table 4. It demonstrates that CL in rat plasma were stable in storage at −20 °C for 30 days and after three cycles of freeze-thaw, and processed samples were stable for 18 h in autosampler. The stabilities of CL were acceptable at the above conditions and would satisfy the requirements of a routine pharmacokinetic study.

### 3.4. Pharmacokinetic and Bioavailability Evaluation

The validated method was used to investigate the pharmacokinetic profiles of CL in rats after single IV and PO administration of pure CL and CL-SFNP, respectively. Mean concentration-time curves of CL are showed in Figure 4. Meanwhile, the major pharmacokinetic parameters estimated by non-compartmental analysis are listed in Table 5; Table 6. For IV administration, it was found that C_max_ and AUC_0-∞_ of CL after CL-SFNP administration were 4414.8 ± 1666.1 ng/mL and 8646.1 ± 1998.9 h*ng/mL respectively, which were significantly (*p* < 0.05) higher than that after pure CL administration (1701.3 ± 170.7 ng/mL and 4697.7 ± 723.0 h*ng/mL, respectively). This implied that CL loaded in silk fibroin nanocapsulation definitely increased its residence time and slowed its elimination in vivo. Similarly, it exhibited significant pharmacokinetics improvement for PO administration. C_max_ (90.5 ± 49.2 ng/mL) and AUC_0-∞_ (1065.5 ± 494.6 h*ng/mL) after CL-SFNP administration were higher than those (35.1 ± 7.9 ng/mL and 441.9 ± 82.6 h*ng/mL, respectively) after pure CL administration. Therefore, CL loaded in silk fibroin nanocapsulation improved its absorption and systemic exposure in vivo. In addition, it was found there were rebound peaks approximately at 2 h in IV, at 2 h and 6 h in PO administration of CL-SFNP, possibly indicating gastric secretion-enteral reabsorption and enterohepaticre recycling may be involved in the pharmacokinetics of CL-SFNP, which needs further investigation. In addition, bioavailability calculation was performed through dosage normalization according to IV dose of pure CL. It was found CL’s mean oral absolute bioavailability (*F*, %) after PO administration of CL-SFNP was 7.56%, more than two times higher than that of 3.14% after PO administration of pure CL. Therefore, bioavailability combining other parameters demonstrated an improvement on pharmacokinetic behavior of CL-loaded silk fibroin nanoparticles.

Recently, improving the drugs’ therapeutic efficiency using biopolymer nanoparticles has been a research focus. Silk fibroin is an excellent biopolymer of amphiphilic chemistry with the features of biocompatibility, biodegradablilty and low immunogenicity, and it can improve the bioavailability and pharmacokinetics of poor dissolution characteristic drugs [30]. Therefore, silk fibroin has been used to load many prospective anti-cancer entities with the purpose of cancer therapy improvement [31,32,33]. However, the in vivo pharmacokinetics investigation of drug loaded silk fibroin nanoparticles was relatively lacking. In this study, we performed pharmacokinetics and bioavailability evaluation of CL-SFNP by developing a reliable LC-MS/MS method of CL determination. The result of improved in vivo pharmacokinetic properties of CL-SFNP, combined with the favorable results from previous in vitro cytotoxicity study against cancer cells, gives hope for the potential of pharmacological activity in vivo of this formulation.

## 4. Conclusions

In conclusions, we developed a simple, sensitive, and reliable LC-MS/MS method to determine CL concentration in rat plasma. In the method, a simple one step liquid–liquid extraction technique using *tert*-butyl methylether as the extract solvent was applied with extraction recovery of about 70%. The method exhibited higher sensitivity with the lower limit of quantification of 0.5 ng/mL using a small plasma volume of 50 μL, and meanwhile it demonstrated reliable accuracy, precision and stability. The developed method was successfully applied to the pharmacokinetic and bioavailability evaluation of CL after IV and PO administration of pure CL and CL-SFNP in rats. The results confirmed the improvement of CL in vivo pharmacokinetics and oral bioavailability upon silk fibroin nanocapsulation. This study provides valuable information for future CL product development.

## Figures and Tables

**Figure 1 molecules-25-03422-f001:**
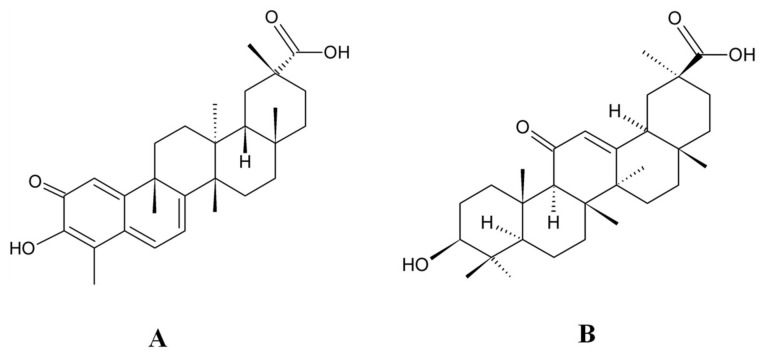
The chemical structure of celastrol (**A**) and 18-α glycyrrhetnic acid (IS) (**B**).

**Figure 2 molecules-25-03422-f002:**
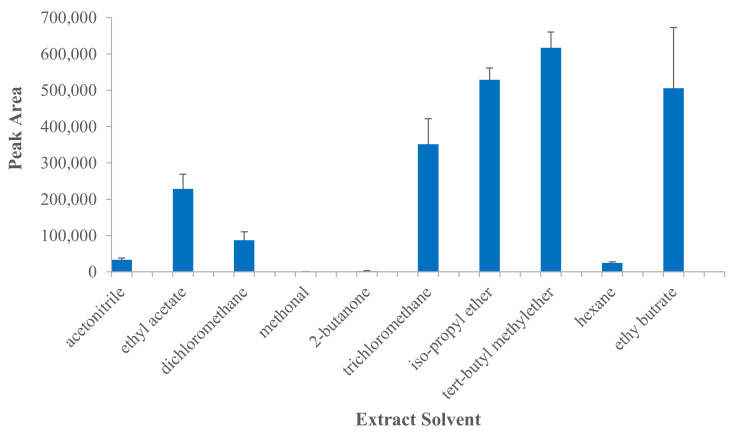
Peak area levels of 100 ng/mL celastrol in rat plasma using different extract solvents (*n* = 3).

**Figure 3 molecules-25-03422-f003:**
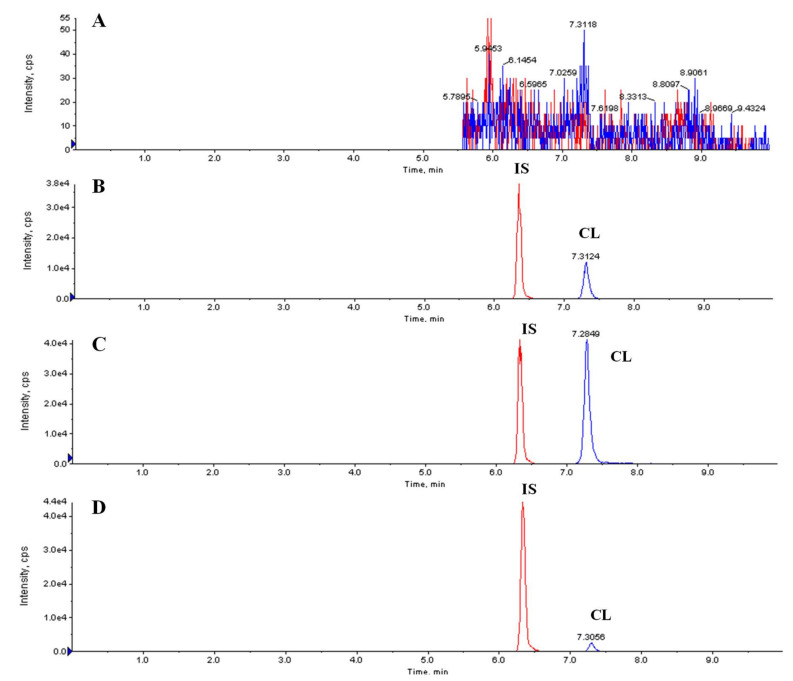
Representative multiple-reaction monitoring (MRM) chromatograms of CL and IS. (**A**) blank plasma; (**B**) blank plasma spiked with CL (100 ng/mL) and IS; (**C**) plasma sample obtained at 0.5 h after intravenous (IV) administration of celastrol-encapsulated silk fibroin nanoparticle (CL-SFNP) suspension; (**D**) plasma sample obtained at 0.5 h after oral (PO) administration of CL-SFNP suspension.

**Figure 4 molecules-25-03422-f004:**
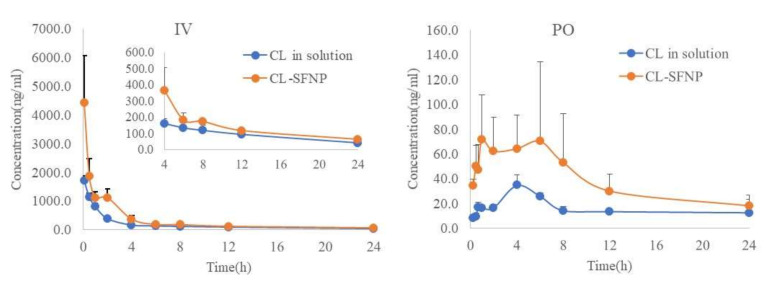
Mean concentration–time curves of CL after IV and PO administration of equivalent CL in pure CL and CL-SFNP (1 mg/kg and 3 mg/kg, respectively) in rats (*n* = 3).

**Table 1 molecules-25-03422-t001:** The peak area ratios (celastrol (CL)/IS) obtained from investigation of different extract conditions for celastrol determination in rat plasma (*n* = 6).

Concentration (ng/mL)	One-Step Extraction with 5 min	One-Step Extraction with 10 min	Two-Step Extraction with Each 5 min
1	0.0035 ± 0.0003	0.0036 ± 0.0002	0.0037 ± 0.0003
25	0.0881 ± 0.0037	0.0858 ± 0.0041	0.0899 ± 0.0039
200	1.0483 ± 0.1025	0.9752 ± 0.0604	1.0361 ± 0.0541

**Table 2 molecules-25-03422-t002:** Precision and accuracy of the assay of celastrol in rat plasma (*n* = 6).

Concentration Spiked (ng/mL)	Intra-Day			Inter-Day		
	Concentration Measured (ng/mL)	Accuracy (%)	Precision (%)	Concentration Measured (ng/mL)	Accuracy (%)	Precision (%)
1	0.91 ± 0.08	91.1	9.1	1.03 ± 0.07	102.6	7.2
25	26.77 ± 2.40	107.1	9.0	24.38 ± 2.85	97.5	11.7
200	220.03 ± 13.15	110.0	6.0	196.76 ± 19.74	98.4	10.0

**Table 3 molecules-25-03422-t003:** The extraction recovery and matrix effect of celastrol in rat plasma (*n* = 6).

Concentration (ng/mL)	Extraction Recovery (%)	Matrix Effect (%)
1	67.0 ± 8.9	101.2 ± 8.8
25	63.5 ± 2.5	87.3 ± 5.9
200	74.7 ± 2.3	98.0 ± 11.0

**Table 4 molecules-25-03422-t004:** The stability of celastrol in rat plasma (*n* = 6).

Concentration Spiked (ng/mL)	In Autosampler after Preparation for 18 h		After Three Freeze-Thaw Cycles		At −20 °C for 30 Days		
	Concentration Measured (ng/mL)	Deviation (%)	Concentration Measured (ng/mL)	Deviation (%)	Concentration Measured (ng/mL)	Deviation (%)	
1	0.88 ± 0.06	−12.4	0.95 ± 0.09	−5.1	0.97 ± 0.08	−3.3	
25	25.44 ± 3.14	1.8	24.28 ± 1.82	−2.9	23.32 ± 2.02	−6.7	
200	181.99 ± 8.21	9.0	180.01 ± 10.37	−10.0	221.64 ± 17.32	10.8	

**Table 5 molecules-25-03422-t005:** Pharmacokinetics parameters of CL after intravenous administration of pure CL and CL-SFNP at 1 mg/kg (*n* = 3).

Parameters	CL in PEG 300	CL-SFNP
*k*_e1_ (h^−1^)	0.0684 ± 0.0092	0.0640 ± 0.0151
*T*_1/2β_ (h)	10.27 ± 1.47	11.27 ± 2.83
*C*_max_ (ng/mL)	1701.3 ± 170.7	4414.8 ± 1666.1 *
*AUC*_0-t_ (h*ng/mL)	4124.3 ± 663.8	7600.4 ± 1658.8 *
*AUC*_0-∞_ (h*ng/mL)	4697.7 ± 723.0	8646.1 ± 1998.9 *
*V*_d_ (mL)	990.3 ± 272.1	544.5 ± 88.7
*Cl* (mL/h)	66.3 ± 12.5	34.6 ± 9.3 *
*MRT*_0-∞_ (h)	9.47 ± 0.77	8.83 ± 2.51

* *p* < 0.05.

**Table 6 molecules-25-03422-t006:** Pharmacokinetics parameters of CL after oral administration of pure CL and CL-SFNP at 3 mg/kg (*n* = 3).

Parameters	CL in PEG 300	CL-SFNP
*T*_1/2_ (h)	12.02 ± 8.32	8.97 ± 2.57
*T*_max_ (h)	4.67 ± 1.15	3.00 ± 2.65
*C*_max_ (ng/mL)	35.1 ± 7.9	90.5 ± 49.2
*AUC*_0-t_ (h*ng/mL)	308.9 ± 45.1	842.9 ± 567.9
*AUC*_0-∞_ (h*ng/mL)	441.9 ± 82.6	1065.5 ± 494.6 *
*V_d_* (mL)	367.1 ± 279.0	261.0 ± 73.2
*Cl* (mL/h)	20.5 ± 1.4	20.2 ± 0.2
*MRT*_0-∞_ (h)	17.01 ± 8.95	13.31 ± 2.67
*F* (%)	3.14 ± 0.59	7.56 ± 3.51 *

* *p* < 0.05.

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
