# Peer review of "Oral Bioavailability Evaluation of Celastrol-Encapsulated Silk Fibroin Nanoparticles Using an Optimized LC-MS/MS Method"

_molecules, 2020, doi:10.3390/molecules25153422_

Round 1

Reviewer 1 Report

Zhan et al. presented the development and validation of the HPLC-MS/MS method for determination of celastrol in rat plasma. The method was used to estimate the bioavailability of celastrol-encapsulated silk fibroin particles in comparison with a simple solution.

The manuscript is very well written, the aim of the study is justified. The experiments were well planned, and all the criteria for bioanalytical method validation were met. The authors thoroughly explained every step of the method optimization. The animal study was undertaken with care, and pharmacokinetics of celastrol was correctly evaluated. I have only a few minor suggestions for the authors:

Lines 64 - 70: It is a rather summary and conclusion of the paper. Please, move it into an appropriate position.
Line 173 - ANOVA instead of ANONA
Figure 3 - This Figure is of low quality, a better resolution is needed.

Author Response

Point 1: Lines 64 - 70: It is a rather summary and conclusion of the paper. Please, move it into an appropriate position.

Response 1: We have moved the content of Line 64-70 to the part of Conclusion and made some minor revision at the conclusion. 

Point 2: Line 173 - ANOVA instead of ANONA

Response 2: We have corrected the word "ANONA" at Line 175 with "ANOVA".

Point 3: Figure 3 - This Figure is of low quality, a better resolution is needed.

Response 3: We have revised Figure 3 after re-organization of these figures.  

Reviewer 2 Report

The article titled “Oral Bioavailability Evaluation of Celastrol-Encapsulated Silk Fibroin Nanoparticles Using an Optimized LC-MS/MS Method” describes the optimization of LC-MS/MS process to analyze the drug, celastrol. Although the article reports an interesting process and written nicely, it lacks enough scientific advancement from the reported techniques. Also, the article is presented in a well-thought rigorously written laboratory report format, rather than as a scientific exploration. So, in its present condition, the authors are encouraged to significantly modify the text, include more experiments and explanations to improve the quality. The article may not suitable for publication in ‘Molecules’ in its present format.

Author Response

Response: Thank you for your comments. We would try to do more better to improve our studies. 

Reviewer 3 Report

The authors report the development of the dosage of celastrol, a natural compound with
anticancer properties. The described method is based on liquid chromography coupled with
mass spectrometry. The method was used to compare the pharmacokinetics of the product
administered in 2 forms, including a nano-encapsulated form
The article is well constructed, easy to read. the analytical method has been adopted according
to international guide lines. The application to the pharmacokinetics of the product in animals
as well as the determination of pharmacokinetic parameters are rigorously carried out.
The article can be accepted with minor modifications listed below:
Material and methods:
Section 2.1. it would be interesting to indicate the structure of the internal standard. this
structure could be added to figure 1
section 2.6.
- The authors must indicate under which number, their protocol has been approved after the
animal ethics committee
- why was CL not administered at the same dose by gastric route and IV route ? is this a DL50
problem? Is the LD 5O of this compound known? was it determined by the authors? Authors
should indicate this LD50 value
section 3.2 The others indicate that they obtain an average extraction yield of 70% with terbutyl methylether. it indicates that this yield is the best of those reported in the literature.
However, line 193, the authors indicate that yields of between 50 and 90 with ethyl acetate have
been reported in the literature, therefore for some, better than that reported by the authors.
Authors should clarify this point
Results: Figure 4 is too small and seems difficult to read. It should be enlarged.
On the other hand, for the IV route, the concentrations after the 5th hour are very less than 1000
ng / mL. we cannot know in which concentration zone they are located. Maybe it would be
desirable to make 2 graphs, one for the first 5 hours of kinetics, the other for the end
Another solution, shown in a box in the same graph, the end of the kinetic with another ordinate
scale

Author Response

Point 1: Material and methods:
Section 2.1. it would be interesting to indicate the structure of the internal standard. this structure could be added to figure 1

Response 1: We have added the structure of internal standard in Figure 1.

Point 2:- The authors must indicate under which number, their protocol has been approved after the animal ethics committee

Response 2: We have added the protocol number of R18IACUC001 approved by IACUC in the revised manuscript.

Point 3:- why was CL not administered at the same dose by gastric route and IV route ? is this a DL50 problem? Is the LD 5O of this compound known? was it determined by the authors? Authors should indicate this LD50 value

Response 3:  Based on information from the literature, the bioavailability of celastrol is very low. So the IV dose was selected to be lower than the oral dose. 1-5 mg/kg is the common range of celastrol doses when given intravenously according to the literature. LD50 of celastrol is not known.

Point 4: section 3.2 The others indicate that they obtain an average extraction yield of 70% with terbutyl methylether. it indicates that this yield is the best of those reported in the literature. However, line 193, the authors indicate that yields of between 50 and 90 with ethyl acetate have been reported in the literature, therefore for some, better than that reported by the authors. Authors should clarify this point

Response 4: In this point, we tried to elaborate the necessary of sample preparation optimization in this study because we found there were chaotic reported results of extraction in the literature, such as from 60% to 90% even using the same extract solvent as ethyl acetate with one-step extraction. We maybe can not indicate 70% is the highest extraction recovery but it is the best result obtained from our detailed sample preparation optimization. But for clarification,  we have deleted the word of "highest" in the revised manuscript. 

Point 5: Results: Figure 4 is too small and seems difficult to read. It should be enlarged.
On the other hand, for the IV route, the concentrations after the 5th hour are very less than 1000 ng / mL. we cannot know in which concentration zone they are located. Maybe it would be desirable to make 2 graphs, one for the first 5 hours of kinetics, the other for the end Another solution, shown in a box in the same graph, the end of the kinetic with another ordinate scale

Response 5: We have revised the Figure 4 with adding an enlarged graph after 4th hour for IV route. 

Reviewer 4 Report

The manuscript entitled "Oral Bioavailability Evaluation of Celastrol-Encapsulated Silk Fibroin Nanoparticles Using an Optimized LC-MS/MS Method" by Zhang et al. describes an improved method for extracting and evaluating the concentration of celestrol (CS), a promising therapeutic agent, in rat blood plasma. The authors demonstrate a more efficient liquid-liquid extraction in an optimized solvent (tert-butylmethyl ether), and evaluate CL levels using an LC-MS-MS method using sample volumes that are low (50 uL) compared to similar studies without sacrificing too much sensitivity. The authors further show that their established method of nano-encapsulating CL in silk fibroin nanoparticles increases the blood plasma concentration and retention of CL in both intravenous and oral delivery methods. While the study is not extremely novel or transformative, it still conveys an important result that proves how encapsulation can improve the bioavailability of these hydrophobic agents.  The manuscript is generally well written and I believe it is suitable for publication in Molecules, but I recommend the following minor concerns are addressed:

Section 2.3. I am confused about the significant figures in this section. How can the authors prepare a quality control sample at CL concentrations of 200.00, 25.00, 1.00, etc. (all with different numbers of sig figs) from a 1 sig-fig stock solution of 1 mg/ml and its dilutions of 500, 200, etc.? Should the stock solution concentration be 1.0000 mg/mL?

Line 197: correct “ethy” to “ethyl”

Figure 3. The low resolution and composition of this figure is unacceptable for publication. It appears as if the chromatograms have been copied from a screen shot and compressed at very low resolution. They are also organized in a sloppy way such that the different rectangles do not have the same aspect ratio and they form a grid where the different cells are not perfectly aligned. Please improve the resolution and organization of this figure.

There appears to be a number of inconsistencies in the references cited. Some of the citations include full page ranges (e.g. 123-124), while others contain partial page ranges (e.g. 123-4). Some references seem to include issue numbers while others do not. At least one reference (#18) is either missing page numbers of the page numbers are in parenthesis. The authors should carefully check their references in make sure they conform to journal standards.

Author Response

Point 1: Section 2.3. I am confused about the significant figures in this section. How can the authors prepare a quality control sample at CL concentrations of 200.00, 25.00, 1.00, etc. (all with different numbers of sig figs) from a 1 sig-fig stock solution of 1 mg/ml and its dilutions of 500, 200, etc.? Should the stock solution concentration be 1.0000 mg/mL?

Response 1: The stock solution concentration should be 1.0000mg/mL. The three QC concentration should be expressed as 200, 25 and 1 ng/mL. The QC solutions were prepared from the stock solution through serial dilution and the dilution method was that the stock solution (1.0000 mg/mL) diluted 100 times to get the solution of 10 µg/mL, and then further serial diluted 50, 8 and 25 times to get the three QC solutions. We have revised all the significant figures in Section 2.3., Section 3.3.3., Table 1, Table 2, Table 3 and Table 4, respectively.     

Point 2: Line 197: correct “ethy” to “ethyl”

Response 2: We have corrected "ethy" to "ethyl" at Line 201 in the revised manuscript.

Point 3: Figure 3. The low resolution and composition of this figure is unacceptable for publication. It appears as if the chromatograms have been copied from a screen shot and compressed at very low resolution. They are also organized in a sloppy way such that the different rectangles do not have the same aspect ratio and they form a grid where the different cells are not perfectly aligned. Please improve the resolution and organization of this figure.

Response 3: We have re-organized the chromatograms and replaced the Figure 3 with a figure of improved resolution.

Point 4:  There appears to be a number of inconsistencies in the references cited. Some of the citations include full page ranges (e.g. 123-124), while others contain partial page ranges (e.g. 123-4). Some references seem to include issue numbers while others do not. At least one reference (#18) is either missing page numbers of the page numbers are in parenthesis. The authors should carefully check their references in make sure they conform to journal standards.

Response 4: We have carefully checked all the references and made relative revisions to unify the cited format.  

Round 2

Reviewer 2 Report

Authors' response towards improving the quality of the manuscript was inadequate.

Author Response

Point 1: Although the article reports an interesting process and written nicely, it lacks enough scientific advancement from the reported techniques. 

Response 1: From the view of celastrol determination, although there were some reported methods (References 18-22), there were some defects such as complicated mobile phase, larger sample volume and lower extraction recovery, which we have addressed in the introduction (Line 51-59). So, we thought the method of celastrol determination could be improved through sample preparation optimization which have not be performed in the reported method.  Although our sample preparation optimization is conventional, we indeed established a simple and reliable method with smaller sample volume and higher extraction recovery than those reported. In addition, our focus was on the pharmacokinetics and bioavailability of novel celastrol formulation with encapsulation using silk fibroin, and it shows encapsulation could improve the bioavailability of celastrol.

Point 2: Also, the article is presented in a well-thought rigorously written laboratory report format, rather than as a scientific exploration.

Response 2: We appreciate your comments. We have addressed the suitable significance of this study in the introduction and added discussion (Line 272-282). 

Round 3

Reviewer 2 Report

Probably be accepted.